



# Lidar observations of atmospheric internal waves in the boundary layer of atmosphere on the coast of Lake Baikal

Viktor A. Banakh, Igor N. Smalikho

V.E. Zuev Institute of Atmospheric Optics SB RAS, Tomsk, Russia

*Correspondence to*: V.A.Banakh (banakh@iao.ru)

**Abstract.** Atmospheric internal waves (AIWs) in the boundary layer of atmosphere have been studied experimentally with the use of Halo Photonics pulsed coherent Doppler wind lidar Stream Line. The measurements were carried out in August 14-28 of 2015 on the western coast of Lake Baikal (52ºN, 105ºE), Russia. The lidar placed at a distance of 340 m from Lake Baikal at a height of 180 m. Probing ranges covered heights from 280 to 1180 m above the level of Lake Baikal.

A total of six AIW occurrences have been revealed. This always happened in the presence of one or two (in 5 of 6 cases) narrow jet streams at heights of approximately 200 and 700 m. The period of oscillations of the wave addend of the wind velocity components in four AIW events was 9 min, and in two other it was approximately 18 and 6.5 min. The amplitude of oscillations of the horizontal wind velocity component was about 1 m/s, while the amplitude of oscillations of the vertical velocity was three times less. In the most cases, internal waves were observed for 45 min (5 wave oscillations with a period

of 9 min). Only one time the AIW lifetime was about four hours.

## 1 Introduction

Atmospheric gravity waves (AGWs) are an important feature of motions present in the atmosphere. They are responsible for transfer of addititious mechanic and thermal energy, which leads to the spatial inhomogeneity and temporal variability of the wind and temperature fields. As AGWs destroy, the released energy causes turbulization of the wind and temperature fields.

Study of the gravity waves is carried out with the help of space images of the cloud fields in the visible and microwave regions (for example, [German, 1985; Xiaofeng et al., 2001]) and radar images of the sea surface (for example, [Spiridonov et al., 1987; Chunchuzov et al., 2000]). Experimental investigations of AGWs in the ionosphere from the scattering of radio waves are carried out by different methods [for example, Benediktov et al., 1997]. The first results of lidar observations of the inertia gravity waves in the stratosphere and mesosphere with the use of the Doppler Rayleigh lidar are reported in

[Baumgarten et al., 2015].

However, the data of AGW observations in the lower atmosphere, in particular, in the atmospheric boundary layer (ABL) are few and far between. As a rule, they are based on the aerological data or on sodar data for the atmospheric lower 300-400 m layer (for example, [Lyulyukin et al., 2015]). At the same time commercially available pulsed coherent Doppler wind lidars (CDWLs) are used actively now in study of the ABL dynamics. Conical scanning by a probing laser beam around the vertical axis



allows reconstruction of the vertical wind profile from wind velocity projections onto the axis of the probing beam estimated from the raw lidar data.

Figure 1 shows the geometry of measurement by pulsed CDWL with the conical scanning by the probing beam around the vertical. During the measurements, the elevation angle $\varphi$ is fixed, while the azimuth angle $\theta = \omega_s t$ of the beam axis

position varies with time $t$ and with the rate $\omega_s$. The recorded lidar signal received from the distance $R_k$ at the azimuth angle

$\theta_n$, where $k, n = 1, 2, 3, \dots$, is then processed to reconstruct the wind vertical profile [Banakh and Smalikho, 2013].

In this paper we present the results of lidar observations of the coastal-mountain lee waves on the coast of Lake Baikal. Lee AIWs (or orographic waves) are one of the types of AGWs, which arise on leeward of obstacles at the stable stratification of an incoming flow [Vel'tishchev and Stepanenko, 2006; Kozhevnikov, 1999]. Experimental investigations of AIWs in the

atmospheric boundary layer of Lake Baikal were carried out with the use of the micropulse low-energy Halo Photonics CDWL Stream Line. These lidars find expanding applications in studies of ABL [Sathe and Mann, 2012; Dinther et al., 2015; Päschke et al., 2015; Smalikho and Banakh, 2015a,b; Smalikho et al., 2015 a,b,c; Vakkari et al., 2015]. The measurements were conducted in August 14-28 of 2015 on the western coast of Lake Baikal (52ºN, 105ºE) at the territory of Baikal Astrophysical Observatory of the Institute of Solar-Terrestrial Physics SB RAS, Russia. The lidar was set at a distance of

340 m from Baikal at a height of 180 m (see Fig. 2).

The processing of all raw data measured by the lidar and analysis of the processed data have revealed several cases of formation of atmospheric internal waves for the period of measurements. Formation of one and often simultaneously two narrow jet streams at heights of the atmospheric boundary layer were observed as well. In all the cases, AIWs were formed in the presence of low-level jet streams.

**2 Observations and analysis**

The measurements were carried out continuously at $\varphi = 60°$ and cover heights from 280 to 1180 m above the Lake Baikal level. The time for one scan around the vertical axis $T_{scan} = 2\pi / \omega_s$ varied and was 2 min, 1 min, and 36 s in dependence on atmospheric conditions. The wind in the atmospheric surface layer during the measurements was mostly directed from the north through the mountainous terrain toward Lake Baikal. Due to forest fires, the atmosphere often contained greater

amounts of aerosol, and, correspondingly, the lidar signal-to-noise ratio was rather high.

Figure 3 shows the results of lidar visualization of the wind field during the longest observations of a gravity wave for about 4 hours starting from 12:00 Local Time on August 23 of 2015. Two jet streams were observed simultaneously at heights of about 250 and 750 m.

Figures 4 and 5 show the vertical profiles at 14:31 LT and temporal profiles at a height of 636.5 m of wind taken from data

in Fig. 3. From these figures, we can clearly see oscillations of the wind speed, direction, and vertical component in both height and time. They are especially marked in the period from 13:30 to 15:30, when the amplitude of oscillations of the wind direction is approximately 45°.



Neglecting the wind turbulence, we use the model of a plane wave for the component of the wind velocity vector $V_\alpha$ (subscript $\alpha = z$ for the vertical component, $\alpha = x$ for the longitudinal component, and $\alpha = y$ for the transverse component) in the form [Vinnichenko et al., 1973]

$$V_\alpha(\mathbf{r},t) = <V_\alpha> + \tilde{V}_\alpha(\mathbf{r},t). \tag{1}$$

In Eq.(1) $\mathbf{r} = \{z,x,y\}$ is the radius vector, $t$ is time, $<V_\alpha>$ and $\tilde{V}_\alpha$ are the regular and wave addends of the $\alpha$-th component of the wind velocity, respectively,

$$\tilde{V}_\alpha(\mathbf{r},t) = A_\alpha(z)\sin\left[\psi_\alpha(\mathbf{r}) - 2\pi t/T_v\right], \tag{2}$$

$A_\alpha$ is the wave amplitude, $\psi_\alpha$ is the wave phase, and $T_v$ is the wave period. If the wind direction coincides with the direction of propagation of the internal wave, then $A_y = 0$, $\psi_x = 2\pi x/\lambda_v$, and $\psi_z = 2\pi x/\lambda_v + \pi/2$. Here, $\lambda_v$ is the

10 wavelength of the wave propagating with the speed $c_v = \lambda_v/T_v$.

Model (1), (2) was applied in the analysis of data in Fig.3 for a height of 766.4 m and 47-min time interval starting from 14:20. From these data, with allowance made for the linear trend, we found the wave addends $\tilde{V}_\alpha(\mathbf{r},t)$ for the three components of the wind velocity vector. In Fig. 6, the solid curve shows the dependence of $\tilde{V}_x$ on $t$. To determine the wave frequency $f_v = 1/T_v$, we have used experimental function $\tilde{V}_x(t)$ and calculated the spectral density, which is depicted in

Fig.7. The obtained spectrum has a peak, from whose position we have determined the frequency $f_v$ to be equal to 0.00185 Hz. Consequently, the wave period is $T_v = 9$ min. Using the least-square fitting of model (2) for $\tilde{V}_x(t)$ to the wave addend of the wind velocity component measured by the lidar (solid curve in Fig. 6), we have determined the phase $\psi_x$ and the amplitude $A_x$. The amplitude of wave addend for the longitudinal component of the wind velocity vector turned out to be 0.96 m/s. The modeled temporal profile $\tilde{V}_x(t)$ calculated by Eq. (2) with the use of experimental values of $A_x$, $\psi_x$, and $T_v$

is shown as a dashed curve in Fig. 6.

Parameters of the wave addend of the vertical wind velocity $\tilde{V}_z(t)$ were found in the same way. The estimates of periods of the internal wave for the longitudinal and vertical components coincided fully ($T_v = 9$ min), amplitude $A_z = 0.3$ m/s is approximately 3 times less than the amplitude of wave addend of the longitudinal component of the wind velocity vector, and $\psi_z - \psi_x = \pi/2$. Since the amplitude $A_y \neq 0$ (see Fig. 3(b) and Fig. 5(b)), the direction of propagation of the internal

wave did not coincide with the wind direction.

Figure 8(a) shows an example of the spatiotemporal distribution of wind velocity, when two jet streams were also observed for 5 hours: one at a height of approximately 200 m, and another at a height of 500 m and higher. Since 5:30 LT, the atmospheric internal wave was observed for about 40 min. According to the data of Fig. 8(b), the period and amplitude of the wave were, respectively, 9 min and 0.9 m/s.



Figure 9 depicts the spatiotemporal distributions of wind and the signal-to-noise ratio in the evening of August 23 for about 45 min. Here we see one jet stream at a height ~730 m and an atmospheric internal wave. In the layer of 100-500 m, the oscillations of the wind speed, direction, and vertical component are accompanied by periodic variations of the signal-to-noise ratio SNR. It is known [Smalikho and Banakh, 2015b] that SNR is proportional to the aerosol backscattering coefficient $\beta_\pi = \rho_c \sigma_\pi$, where $\rho_c$ is the concentration of atmospheric aerosol, and $\sigma_\pi$ is the mean aerosol backscattering cross section. If we assume that during the measurement time (45 min) SNR does not vary due to turbulent pulsations of the refractive index of air and $\sigma_\pi$ remains unchanged, then variations of the aerosol concentration can be estimated from the lidar data as SNR variations.

We used the data of Fig. 9(d) and calculated the relative variations of the aerosol concentration $\eta_c(t) = \rho_c(t)/ <\rho_c>_t -1$, as

$$\eta_c(t) = \text{SNR}(t)/ <\text{SNR}>_t -1, \qquad (3)$$

where the operator $<...>_t$ denotes the time averaging for the period of 45 min.

Since the SNR oscillates within the height range 100-500 m in Fig. 9(d), it is evident, that aerosol concentration should vary with time too. These aerosol concentration (SNR) variations can be caused by oscillations of the vertical component of the wind velocity vector, whose amplitude is relatively high. To test it, we compared $V_z(t)$ with $\eta_c(t)$.

Figure 10 shows the temporal profiles $V_z(t)$ and $\eta_c(t)$ obtained from the data depicted in Fig. 9 for a height of 220.8 m. From the analysis of the curve in Fig.10(a), it follows that the period of oscillations $T_v$ of the vertical component of wind velocity is 6.5 min. The same period of oscillations is also observed for other components of the wind vector, whose phase is shifted by 90° about the phase of $\tilde{V}_z(t)$. According to Fig. 10(b), $\eta_c(t)$ is characterized not only by periodic variations with time, but also by nonstationarity within the considered time interval. It follows from the rough estimates that the period of oscillations of the aerosol concentration is close to $T_v = 6.5$ min, while the phase is shifted from $\tilde{V}_z(t)$ by about 90°.

In addition to these three cases of AIW occurence, we succeeded in observation of this phenomena three times more for the period of measurements. Thus, on August 25 in the predawn time (04:30–05:06), two jet streams and AIW with the period $T_v \approx 9$ min and the amplitude $A_x \approx 1$ m/s at a height of 402.7 m were observed in the atmospheric boundary layer. The next day (August 26 of 2015), the internal wave with the period $T_v \approx 18$ min and the amplitude $A_x \approx 0.7$ m/s at the same height 402.7 m, passed from 16:22 to 19:00 LT. In the same day, the AIW with the halved period ($T_v \approx 9$ min) and the amplitude $A_x \approx 0.4$ m/s at the height 402.7 m was observed 50 min later from 19:50 to 20:35 LT.

**3 Summary**

Thus, the results of the experimental campaign in the coastal zone of Lake Baikal in August of 2015 show that the raw data of measurements by the Stream Line lidar allow us to visualize the spatiotemporal structure of the wind field in the



atmospheric boundary layer and reveal the presence of low-level jet streams and atmospheric internal waves. The distinguishing feature of the atmospheric conditions of the Lake Baikal is occurrence the stable thermal stratification in the ABL during the day tame. The low level jet streams were observed by day and night while none of the AIWs events was in the night time.

A total of six cases of AIW formation have been revealed, which always occurred in presence of one or two (in 5 of 6 cases) narrow jet streams at heights of about 200 and 700 m. When two jet streams were formed, the period of oscillations of the wave addend of the wind vector components was 9 min. In only one case it was about 18 min. In presence of a single jet stream (at a height of 730 m), the period of oscillations of the wind vector components during AIW was about 6.5 min. The amplitude of oscillations of the horizontal wind components most often was about 1 m/s, while the amplitude of oscillations

of the vertical velocity was tree times less. In the most cases, the internal waves were observed for 45 min (5 oscillations with the period $T_v$ = 9 min). Only once the lifetime of the atmospheric internal wave was about 4 hours.

### Acknowledgments

The authors are thankful to A.V. Falits and A.A. Sukharev for the measurements. The authors are grateful to the Institute of Solar-Terrestrial Physics SB RAS for the possibility of using the territory of the Baikal Observatory for the measurements.

This study was supported by the Russian Scientific Foundation for Maintenance and Development, Project No. 14-17-00386.

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



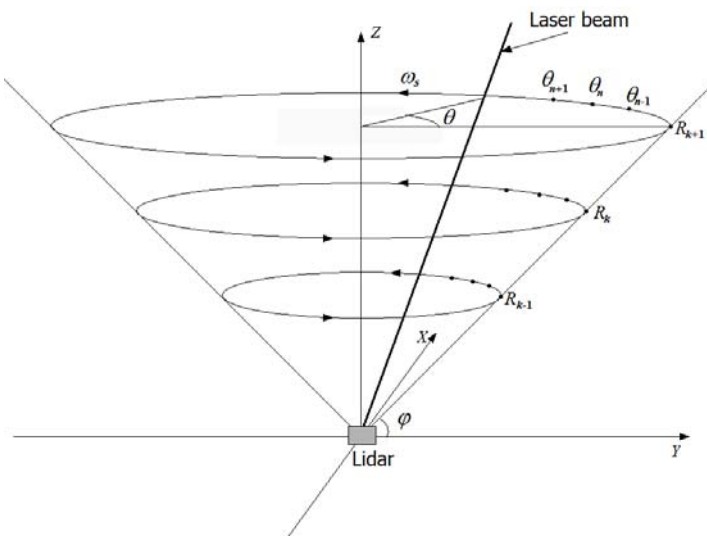

**Figure 1:** Geometry of measurement by a pulsed coherent Doppler lidar with the conical scanning by the laser beam.

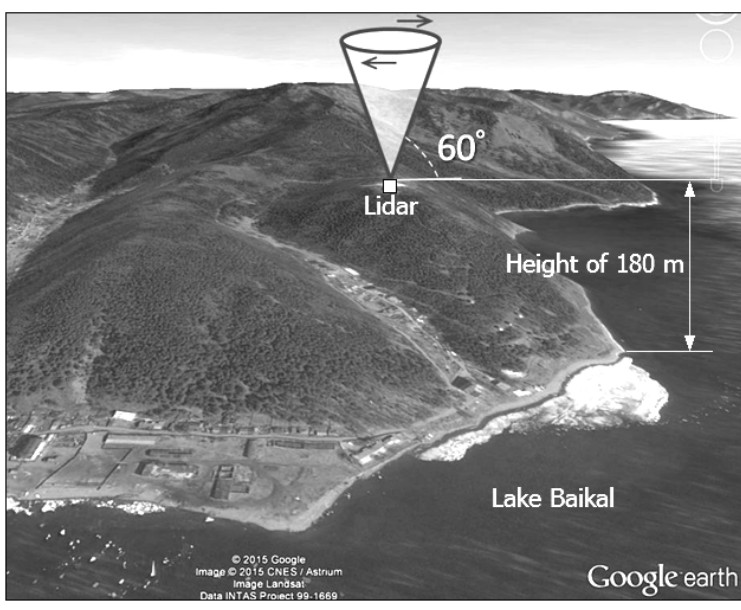

**Figure 2:** Map of lidar wind measurements in August 14-28 of 2015.





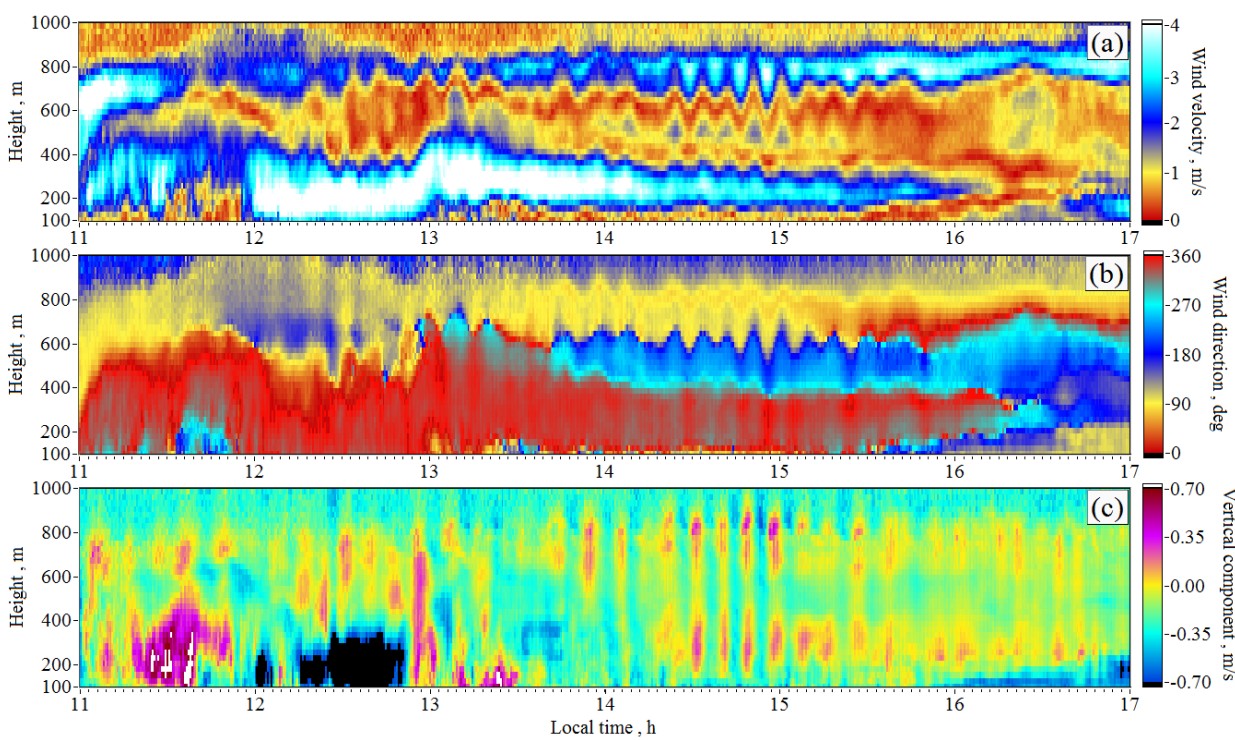

**Figure 3:** Spatiotemporal distributions of the wind speed (a), the wind direction angle (b), and the vertical component of the wind vector (c) obtained from measurements with the Stream Line lidar on August 23 of 2015. The height is given relatively to lidar position height.



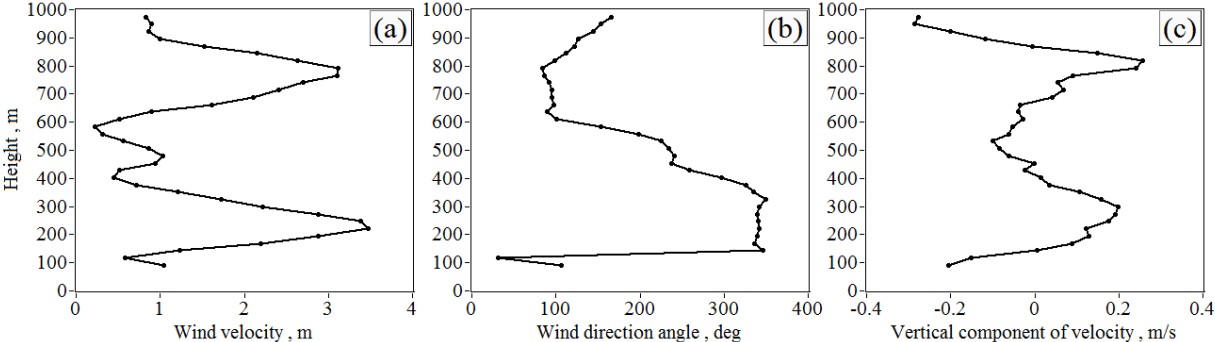

**Figure 4:** Vertical profiles of the wind speed (a), the wind direction angle (b), and the vertical component of the wind vector (c) taken

10      from the data of Fig.3 (these profiles were measured at 14:31 LT).





**Figure 5:** Temporal profiles of the wind speed (a), the wind direction angle (b), and the vertical component of the wind velocity (c) taken from the data of Fig. 3 (measurement height of 636.5 m).





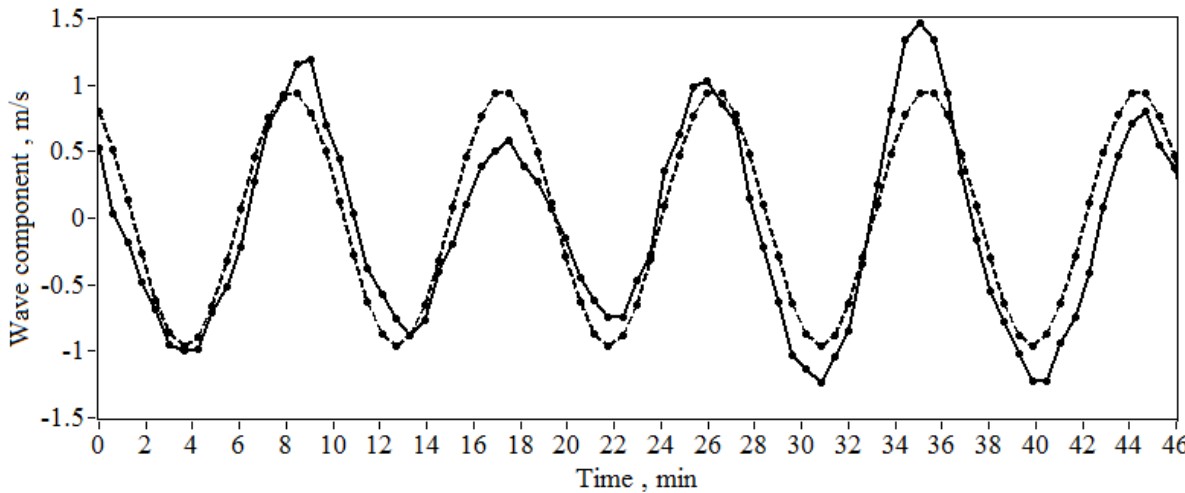

**Figure 6:** Time dependence of the wave addend of the longitudinal wind velocity: (solid curve) measurements by the Stream Line lidar starting from 14:20 LT on August 23 of 2015 at a height of 766.4 m (the data of Fig.3(a) were used); (dashed curve) result of least-square

10   fitting of sine-wave dependence (2) for the wave addend $\tilde{V}_x(t)$ to the measured data  shown by the solid curve.



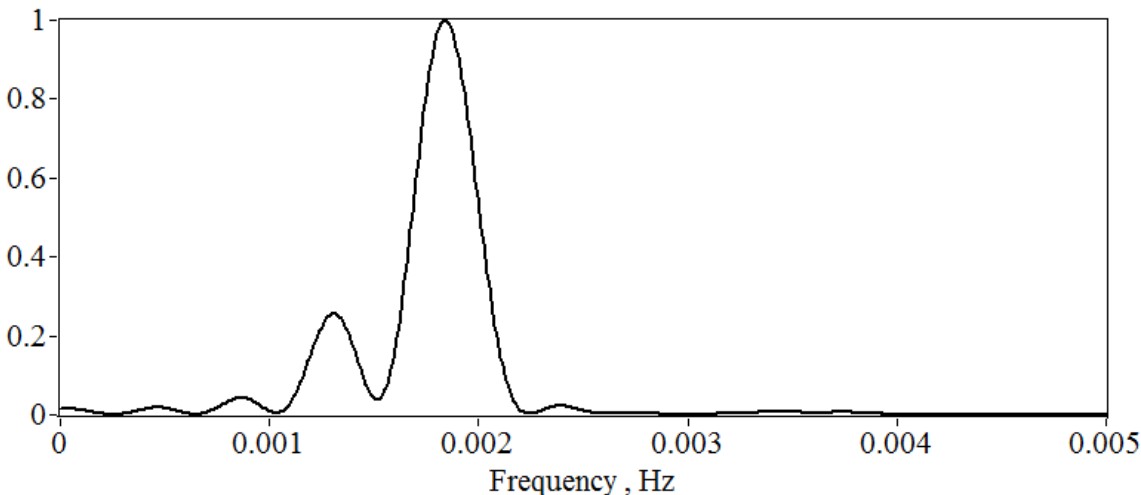

**Figure 7:** Normalized spectrum of the wave addend of wind velocity calculated from the data shown by the solid curve in Fig. 6.





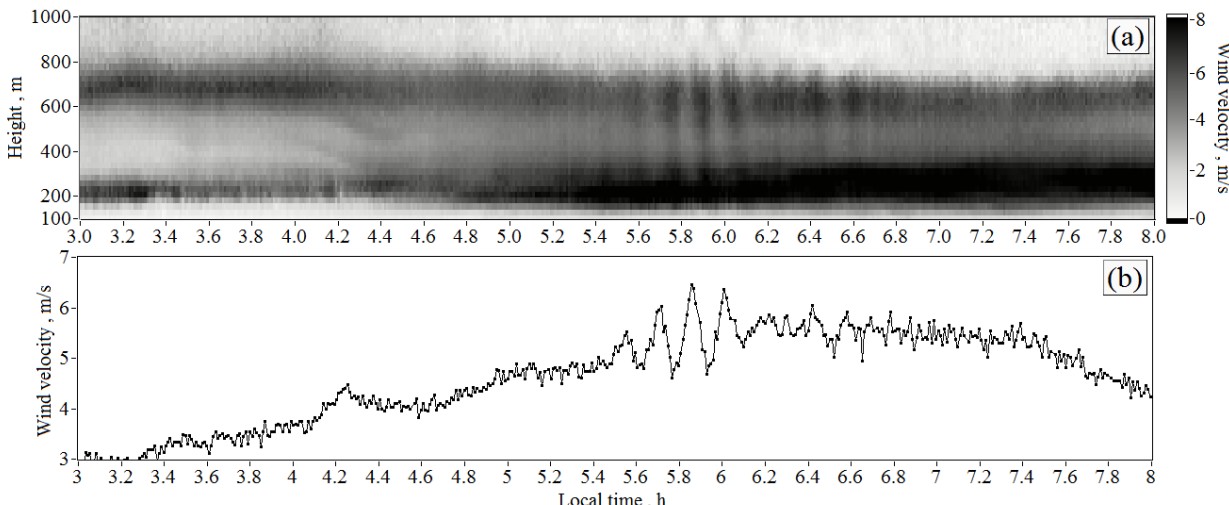

**Figure 8:** Spatiotemporal distribution of the wind velocity (a) and the time profile of the wind velocity at a height of 532.6 m (b) obtained from measurements by the Stream Line lidar on August 19 of 2015.





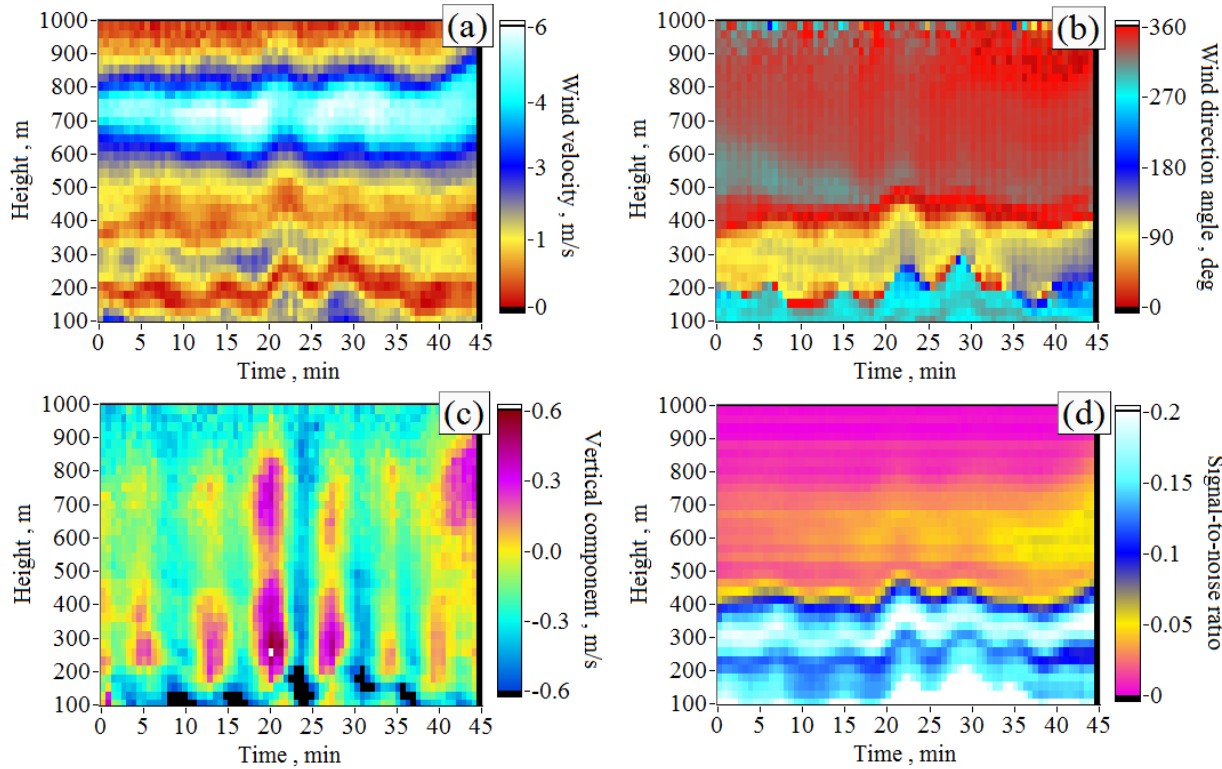

**Figure 9:** Spatiotemporal distributions of the wind speed (a), wind direction angle (b), vertical component of the wind vector (c), and signal-to-noise ratio (d) obtained from measurements of the Stream Line lidar on August 23 of 2015 starting from 19:24 LT.





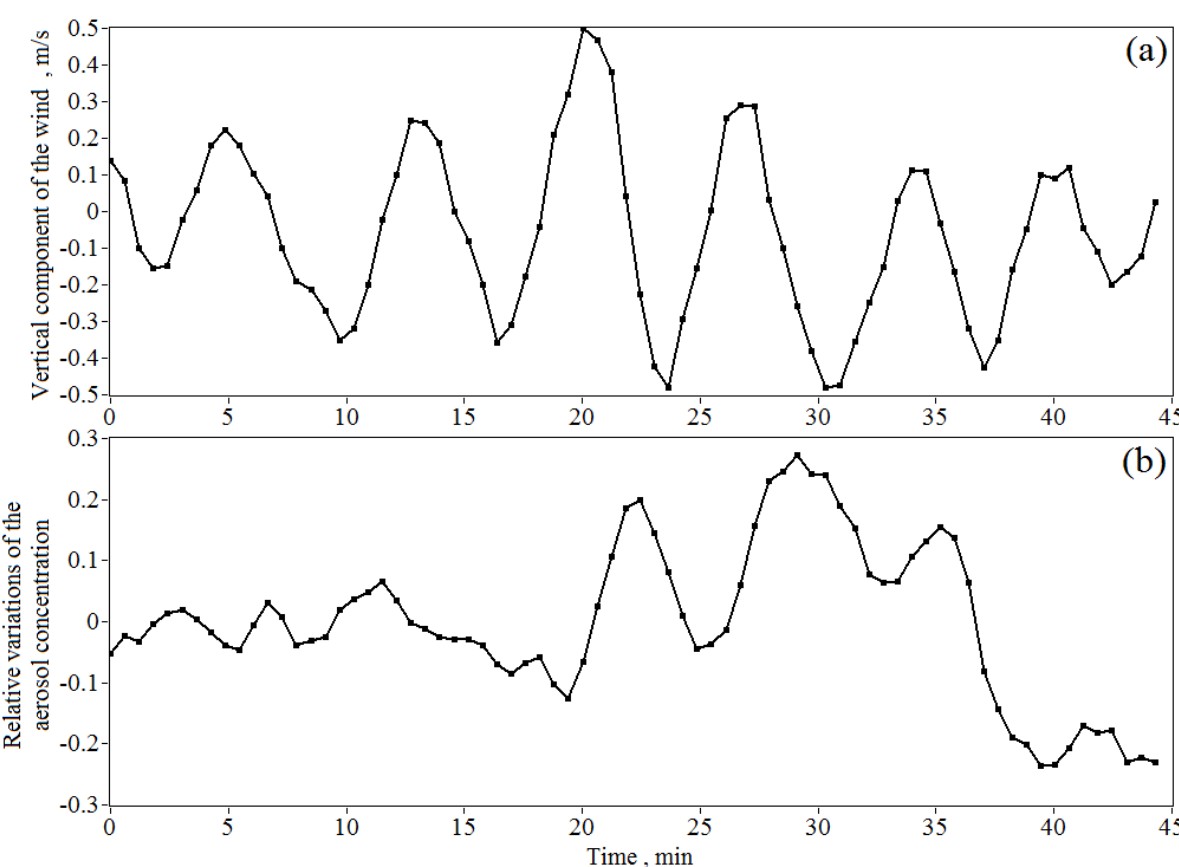

**Figure 10:** Time dependence of the vertical component of the wind vector (a) and relative variations of the atmospheric aerosol concentration $\eta_c(t)$ (b) obtained from the data depicted in Fig. 9(c, d) at a height of 220.8 m.

