# Peer review of "Lidar observations of atmospheric internal waves in the boundary layer of atmosphere on the coast of Lake Baikal"

_Atmospheric Measurement Techniques, 2016_

## Referee Comment (RC1) · Anonymous Referee #2 · 12 Jul 2016

This paper presents a few case studies of atmospheric internal waves near the shore of Lake Baikal as well as methodology to infer some wave properties (amplitude, frequency) from the wind profiles obtained with a scanning Doppler lidar.

I find the presented case studies and methodology interesting enough to merit publication in AMT when following comments have been addressed.

Major comments

First of all, the structure of this manuscript is unclear. Methods are discussed partly in chapter 1 (introduction) and partly in chapter 2 (observations and analysis), which makes it unnecessarily complicated to follow what has been done. Furthermore, ad-

ditional details need to be given on the methods and measurements. Please add one more chapter on methods (separate from results) containing detailed description of the instrument, scanning protocols, data processing, the site and the calculations used in inferring the wave characteristics. Please see detailed comments below for more specific questions.

My second major comment is that the analysis on the variation of aerosol concentration is neglecting several sources of uncertainty. As it seems to me that this section does not contribute significantly to the conclusions of this paper, I recommend the authors to leave it out. If, however, this section is included, all sources of uncertainty in the analysis have to be accounted for. Please see detailed comments below for more specific comments.

Finally, the results are not discussed with respect to previous studies. Examples of a few previous studies on atmospheric gravity waves (AGW) with a couple of different techniques are listed, but no comparison is made to the characteristics of previous AGW observations or theoretical work. As a consequence the implications of this study (with respect to e.g. NWP, wind power or transport of momentum, energy and trace compounds) are not discussed at all. Please summarize current knowledge on atmospheric internal waves in enough detail to place this work in its context and discuss the differences and similarities of the wave properties from this study with previous observations. For instance, the review by Sun et al. (2015) and the recent studies by Roman-Cascon et al. (2015) and Chouza et al. (2016) might provide a starting point.

Detailed comments

Page 2, line 10: As there are a couple of different versions of the Halo Streamline lidar, the manuscript has to contain at least the following details of the system in use: wavelength, pulse repetition rate, Nyquist velocity, sampling frequency, velocity resolution, minimum range, range resolution, points per range gate, pulse duration, lens diameter, lens divergence, telescope, integration time per ray. Integration time per ray seems to

be changing during the campaign (page 2, line 22) and should be given at least for each case study presented here. Also a reference to a detailed description of the Halo lidar (Pearson et al., 2009) would be good to include.

Page 2, line 21-22: It seems that the wind profiles in this study were obtained with a vertical azimuth display (VAD) scan at 60 degree elevation angle. If the VAD scan was carried out by stepping the azimuth angle (as page 2, line 4-6 suggests), please give the number of azimuthal angles included in each scan. Please indicate if the lidar was scheduled to do only VAD scans, or if other scan types were included in the measurement protocol.

Please give details on the Doppler lidar data processing. Was some constant threshold applied to SNR to discard range gates with too low signal or were low-signal measurements filtered in some other way? Was the SNR checked for the artefact described by Manninen et al. (2016)? How were missing radial wind observations (e.g. discarded due to too low SNR) handled in reconstructing the wind profile (c.f. Päschke et al., 2016)? Please discuss also the uncertainty in the obtained horizontal and vertical winds (see e.g. Päschke et al., 2016).

Page 2, line 12-14 : "The measurements were conducted in August 14-28 of 2015 on the western coast of Lake Baikal (52°N, 105°E) at the territory of Baikal Astrophysical Observatory of the Institute of Solar-Terrestrial Physics SB RAS, Russia." Please give the measurement site location with better accuracy, 0.01 degree latitude/longitude (or better) would be suitable. As the manuscript deals with "coastal-mountain lee waves" (page 2, line 7), please present a topographic map of the area of interest (with measurement location indicated) so that the reader can assess the role of terrain height on the observations. How far off from a mountain top are these waves observed? Is it the same mountain for each wave case? How did you conclude that the waves reported here are mountain lee waves and not, for instance, generated by jets or fronts (e.g. Plougonven and Zhang, 2014)?
Page 3, lines 9-15: "Model (1), (2) was applied in the analysis of data in Fig.3 for a height of 766.4 m and 47-min time interval starting from 14:20. From these data, with allowance made for the linear trend, we found the wave addends Va(r,t) for the three components of the wind velocity vector. In Fig. 6, the solid curve shows the dependence of Vx on t. To determine the wave frequency fv =1/Tv, we have used experimental function Vx(t) and calculated the spectral density, which is depicted in Fig.7." Please state exactly how the model was applied to the data. For instance:

- Is <Va> average (mean or median?) of the u,v,w retrieved from the VAD over the wave period? If yes, was this period determined visually or with some other method?

- What do you mean by "with allowance made for the linear trend" – which parameter has a linear trend and how is it taken into account? How do eq. (1) and (2) look like with this linear trend included?

- How were the time-independent parameters in eq. (2) determined – based on a least squares fit?

- How was the spectral density of Vx(t) in Fig. 7 calculated?

Please provide also some measure of uncertainty for the parameters in eq. (2). For instance bootstrap might be applicable.

Page 4, lines 4-6: "It is known [Smalikho and Banakh, 2015b] that SNR is proportional to the aerosol backscattering coefficient beta = rho sigma, where rho is the concentration of atmospheric aerosol, and sigma is the mean aerosol backscattering cross section." SNR does depend on the aerosol backscattering cross section and concentration, but it depends also on the extinction along the path of the transmitted light (Halo output is attenuated backscatter) and the lidar telescope focus. Furthermore, backscattering cross section depends strongly on relative humidity because of hygroscopic growth of the particles. If this section is included in the final paper the uncertainties from these sources need to be quantified. Please see also Engelmann et al. (2008) for further

[Figure]

Interactive
comment

discussion on aerosol flux measurements using lidar technology.

Page 4, lines 6-8: "If we assume that during the measurement time (45 min) SNR does not vary due to turbulent pulsations of the refractive index of air and sigma remains unchanged, then variations of the aerosol concentration can be estimated from the lidar data as SNR variations." This is probably a good assumption in a well-mixed boundary layer (c.f. Engelmann et al., 2008). However, in this case the aerosol layers are not mixed and the measurement at a constant height can represent quite different aerosol characteristics especially if there was (fresh?) biomass burning smoke present (page 2, line 24). In fact, for the 220 m elevation presented in Fig. 10, the highest SNR seem to coincide with 270 degree wind direction, while lower SNR values are observed for 0-90 degree wind direction (see Fig. 9b). Can you assume that aerosol from opposite wind directions has unchanged optical properties? If not, how does this affect the uncertainty in eq. (3)?

Page 4, line 12-14: "Since the SNR oscillates within the height range 100-500 m in Fig. 9(d), it is evident, that aerosol concentration should vary with time too. These aerosol concentration (SNR) variations can be caused by oscillations of the vertical component of the wind velocity vector, whose amplitude is relatively high." Please see previous comments: SNR is not equal to aerosol concentration.

Minor comments

Please check the language carefully once more.

Page 1, line 8-9: Please specify if height is above ground level or something else

Page 1, line 21: "Xiaofeng et al., 2001" is not included in References

Page 2, line 4-5: "During the measurements, the elevation angle phi is fixed, while the azimuth angle theta = omega t of the beam axis position varies with time t and with the rate omega." To me this statement suggests that the VAD was operated with "continuous wave" method, though the Halo lidar is more commonly operated by stepping

from one azimuth angle to the next, in which case scan duration is determined by the integration time and number of azimuthal directions included in the scan. Please clarify how the VAD scan was operated.

Page 2, line 6: "k,n=1,2,3,..." Please give the number of range gates and azimuth angles used in this study.

Page 2, line 7-9: "Lee AIWs (or orographic waves) are one of the types of AGWs, which arise on leeward of obstacles at the stable stratification of an incoming flow [Vel'tishchev and Stepanenko, 2006; Kozhevnikov, 1999]." There are many earlier studies on lee waves. Some early references can be found e.g. in Sun et al. (2015) and in Makarenko and Maltseva (2011).

Page 2, line 15: Is the height above Lake Baikal level?

Page 2, line 23: "The wind in the atmospheric surface layer during the measurements was mostly directed from the north through the mountainous terrain toward Lake Baikal." What height above ground is "the atmospheric surface layer"? Is this from the Doppler lidar or some surface anemometer?

Page 3, line 1: "Neglecting the wind turbulence, we use the model of a plane wave for the component of the wind velocity vector" Please discuss the uncertainty in neglecting turbulence.

Page 3, line 5: What is the coordinate system for the radius vector?

Page 3, line 8-9: "If the wind direction coincides with the direction of propagation of the internal wave, ..." How is the wavelength determined if the wind direction does not coincide with the direction of propagation of the internal wave?

Page 3, line 24-25: "Since the amplitude A is not 0 (see Fig. 3(b) and Fig. 5(b)), the direction of propagation of the internal wave did not coincide with the wind direction." How large was the angle between wind direction and the direction of propagation of the internal wave (for each wave case)? How does this compare with previous observations?

Page 3, line 28-29: "According to the data of Fig. 8(b), the period and amplitude of the wave were, respectively, 9 min and 0.9 m/s." Is this for horizontal wind speed?

Page 4, line 4-5: "It is known [Smalikho and Banakh, 2015b] that SN is proportional to the aerosol backscattering coefficient" Here a reference to e.g. the instrument description (Pearson et al., 2009) might be more appropriate.

Page 4, line 22: Is time local time?

Page 5, line 5-6: "A total of six cases of AIW formation have been revealed, which always occurred in presence of one or two (in 5 of 6 cases) narrow jet streams at heights of about 200 and 700 m." I find interesting the observation of two simultaneous jet streams. Are there previous observations of such cases from comparable environments? Can you comment on the possible forcing mechanisms of the jet streams?

Figure 1. It seems a little unusual to me to define VAD with a left-handed axis. Is this the axis used for eq. (1) and (2)?

Figure 8. Please use the same colour map as for other wind speed plots.

References

Chouza, F., Reitebuch, O., Jähn, M., Rahm, S. and Weinzierl, B.: Vertical wind retrieved by airborne lidar and analysis of island induced gravity waves in combination with numerical models and in situ particle measurements, Atmos. Chem. Phys., 16(7), 4675–4692, doi:10.5194/acp-16-4675-2016, 2016.

Engelmann, R., Wandinger, U., Ansmann, A., Müller, D., Žeromskis, E., Althausen, D. and Wehner, B.: Lidar Observations of the Vertical Aerosol Flux in the Planetary Boundary Layer, Journal of Atmospheric and Oceanic Technology, 25(8), 1296–1306, 2008.

Makarenko, N. I. and Maltseva, J. L.: Interference of lee waves over mountain ranges,

Nat. Hazards Earth Syst. Sci., 11(1), 27–32, doi:10.5194/nhess-11-27-2011, 2011.

Manninen, A. J., O'Connor, E. J., Vakkari, V. and Petäjä, T.: A generalised background correction algorithm for a Halo Doppler lidar and its application to data from Finland, Atmos. Meas. Tech., 9(2), 817–827, doi:10.5194/amt-9-817-2016, 2016.

Plougonven, R., and Zhang, F.: Internal gravity waves from atmospheric jets and fronts, Rev. Geophys., 52, doi:10.1002/2012RG000419, 2014.

Pearson, G., Davies, F. and Collier, C.: An Analysis of the Performance of the UFAM Pulsed Doppler Lidar for Observing the Boundary Layer, J. Atmos. Oceanic Technol., 26(2), 240–250, doi:10.1175/2008JTECHA1128.1, 2009.

Päschke, E., Leinweber, R., and Lehmann, V.: An assessment of the performance of a 1.5 $\mu$m Doppler lidar for operational vertical wind profiling based on a 1-year trial, Atmos. Meas. Tech., 8, 2251–2266, doi:10.5194/amt-8-2251-2015, 2015.

Román-Cascón, C., Yagüe, C., Mahrt, L., Sastre, M., Steeneveld, G.-J., Pardyjak, E., van de Boer, A. and Hartogensis, O.: Interactions among drainage flows, gravity waves and turbulence: a BLLAST case study, Atmos. Chem. Phys., 15(15), 9031–9047, doi:10.5194/acp-15-9031-2015, 2015.

Sun, J., Nappo, C. J., Mahrt, L., Belušić, D., Grisogono, B., Stauffer, D. R., Pulido, M., Staquet, C., Jiang, Q., Pouquet, A., Yagüe, C., Galperin, B., Smith, R. B., Finnigan, J. J., Mayor, S. D., Svensson, G., Grachev, A. A. and Neff, W. D.: Review of wave-turbulence interactions in the stable atmospheric boundary layer, Reviews of Geophysics, 53(3), 2015RG000487, doi:10.1002/2015RG000487, 2015.

---

## Referee Comment (RC2) · Anonymous Referee #1 · 18 Jul 2016

The paper shows the ability of the Doppler lidar to visualize the spatiotemporal structure of the wind field in the atmospheric boundary layer, and reveal the presence of low-level jet streams and atmospheric internal waves. Six atmospheric internal wave (AIW) occurrences were observed using Halo Photonics Doppler wind lidar measurements. Results of the wind flow analysis along with an explanation of the approach to determine wave frequency, phase and amplitude, were described in the paper. Because of the importance of the AGWs for understanding atmospheric vertical energy and momentum transfer and improvement of model parameterization, information presented in the paper deserves attention in scientific literature.

The manuscript falls into the scope of AMT and provides scientifically sufficient results.

[Figure]

Therefore my recommendation is to accept this manuscript for publication in AMTD after revisions.

Major revisions The paper requires some clarification in the description and the order of cases shown. For example, Figs 3-7 describe an observed case during Aug. 23, then Fig 8 shows the case observed on Aug. 19. Figs 9-10 show another case on Aug. 23. Not clear of the logic behind this jumping from one day to another and coming back to the first day again. It would be easier to read the paper if the results are organized by either the date or time of the observed events, or at least by the period of oscillations of waves. Results presented in Figs 5 and 6 are confusing. Figure 5 showed detailed analysis and temporal profiles at "a height of 636.5 m of wind taken from data in Fig. 3". The following sentence states that: "They [wave oscillations] are especially marked in the period from 13:30 to 15:30." However, Fig. 6 shows results when models 1 and 2 were "applied in the analysis of data ….for a height of 766.4 m and 47-min time interval starting from 14:20." Why do not apply models first to data as in Fig. 5 and then repeat the analysis of data at the other (766.4 m) height and "47-min time interval"? The authors may provide the range of the lidar measurement uncertainty since the values of some wave parameters are very low. They also may reference more similar studies using Doppler lidars such as Yansen Wang et al, 2013, JARS, v.7, "Investigation of nocturnal low-level jet–generated gravity waves over Oklahoma City during morning boundary layer transition period using Doppler wind lidar data" In addition, description of an agreement between presented results with previous studies would only strengthen the paper.

Minor revisions

Abstract Line 9 "jet streams at heights of….700 m." Are these heights above ground level (AGL) at the lidar location or above the surface (ASL) of the lake? Better to state at the beginning what heights (AGL or ASL) were used throughout the paper.

Introduction Line 17 Indent the sentence "Atmospheric. . . Line 25 "[Baumgarten et al.,

2015]". Consider mentioning the paper of Newsom et al, 2003. They also used the Doppler lidar measurements over a flat terrain to characterize the wave field, its interaction with the mean flow, and its role in turbulence generation.

Page 2 Line 15 "340 m from Baikal ". From what point on the lake coast was the distance counted? Indicate this distance on Figure 2. Lines 23-24 "The wind in the atmospheric surface layer during the measurements was mostly directed from the north through the mountainous terrain toward Lake Baikal". The sentence arises the following questions: What is your definition of the atmospheric surface layer here? How far the second mountain hill was from the lidar location (Fig.2)? May it influence the measurements of the northerly winds? Line 28 "250 and 750 m ". You may add AGL Line 31 "They are especially marked in the period from 13:30 to 15:30". Suggest to use the word "evident" instead of "marked"

Page 3 Line 11 "Model (1), (2) was applied." Rewrite as "Models (1) and (2) were applied..."

Summary Page 5, Line 2 "day tame", change to "day time" Suggest to rewrite the next sentence "The low level jet streams were observed by day and night while none of the AIWs events was in the night time," as follows: "The low level jet streams were observed during day and night times while none of the AIWs events were observed in the night time"

---

## Author Comment (AC1) · 19 Sep 2016

Viktor A. Banakh and Igor N. Smalikho

banakh@iao.ru

<Responses for the reviewers of the manuscript>

<We thank very much the reviewers for their time and efforts, thoughtful and very useful comments. We have incorporated most of their suggested revisions as indicated below.>

Anonymous Referee #1

The paper shows the ability of the Doppler lidar to visualize the spatiotemporal structure of the wind field in the atmospheric boundary layer, and reveal the presence of low-level jet streams and atmospheric internal waves. Six atmospheric internal wave

(AIW) occurrences were observed using Halo Photonics Doppler wind lidar measurements. Results of the wind flow analysis along with an explanation of the approach to determine wave frequency, phase and amplitude, were described in the paper. Because of the importance of the AGWs for understanding atmospheric vertical energy and momentum transfer and improvement of model parameterization, information presented in the paper deserves attention in scientific literature. The manuscript falls into the scope of AMT and provides scientifically sufficient results. Therefore my recommendation is to accept this manuscript for publication in AMTD after revisions.

Major revisions

The paper requires some clarification in the description and the order of cases shown. For example, Figs 3-7 describe an observed case during Aug. 23, then Fig 8 shows the case observed on Aug. 19. Figs 9-10 show another case on Aug. 23. Not clear of the logic behind this jumping from one day to another and coming back to the first day again. It would be easier to read the paper if the results are organized by either the date or time of the observed events, or at least by the period of oscillations of waves.

<Figure 9 (Figure 11 in the revised manuscript) has been incorrectly specified day of measurement (should be Aug. 14 instead of Aug. 23). In the beginning we present result obtained from lidar data measured during most long-time presence of the gravity wave in the atmosphere, when wind velocity was rather small (Figure 5 in the revised manuscript). Figure 10 (in revised manuscript) shows gravity wave at relatively strong wind. In 5 cases of 6 the gravity waves were at presence of 2 low-level jet streams. Figures 11 and 12 show oscillations of wind and backscatter coefficient (SNR) at presence of one jet stream. Page 6, line 27: "Figures 5-7 illustrate the appearance . . . is 1 – 2.5 m/s." has been added. Page 6, line 29: "when the averaged wind velocity was about 5.5 m/s" has been added. (page numeration of the revised manuscript)>

Results presented in Figs 5 and 6 are confusing. Figure 5 showed detailed analysis and temporal profiles at "a height of 636.5 m of wind taken from data in Fig. 3". The

following sentence states that: "They [wave oscillations] are especially marked in the period from 13:30 to 15:30." However, Fig. 6 shows results when models 1 and 2 were "applied in the analysis of data . . ..for a height of 766.4 m and 47-min time interval starting from 14:20." Why do not apply models first to data as in Fig. 5 and then repeat the analysis of data at the other (766.4 m) height and "47-min time interval"?

<Figure 5 (Figure 7 in the revised manuscript) shows result for height of 636.5 m (below the upper jet stream), when amplitude of wind direction oscillations is maximal. Figure 6 (Figure 8 in revised manuscript) shows result for height of 766.4 m (inside the upper jet stream), when amplitude of wind velocity oscillation is maximal. Page 5, line 14: "Direction of first jet stream (at height of 250 m) is from North to South and direction of other jet stream is from East to West." has been added. Page 5, line 19: "maximal and equal to" has been added. Page 6, lines 1-2: "(inside the upper jet stream)" and ", when the amplitude of wind velocity oscillations was maximal" have been added.>

The authors may provide the range of the lidar measurement uncertainty since the values of some wave parameters are very low. They also may reference more similar studies using Doppler lidars such as Yansen Wang et al, 2013, JARS, v.7, "Investigation of nocturnal low-level jet–generated gravity waves over Oklahoma City during morning boundary layer transition period using Doppler wind lidar data" In addition, description of an agreement between presented results with previous studies would only strengthen the paper.

<Page 6, lines 17-26: "To estimate turbulence strength . . . error is approximately 3 times less." has been added. References (Newsom and Banta, 2003), (Wang et al., 2013), and (Chousa et al., 2016) have been included.>

Minor revisions

Abstract Line 9 "jet streams at heights of. . ..700 m." Are these heights above ground level (AGL) at the lidar location or above the surface (ASL) of the lake? Better to state at the beginning what heights (AGL or ASL) were used throughout the paper.

<Page 1, line 11:"above ground level at the lidar location" has been added and abbreviation "AGL" for "above ground level" is used throughout the revised manuscript.>

Introduction Line 17 Indent the sentence "Atmospheric. . . Line 25 "[Baumgarten et al., 2015]". Consider mentioning the paper of Newsom et al, 2003. They also used the Doppler lidar measurements over a flat terrain to characterize the wave ïnËĞAËZeld, its interaction with the mean ïnËĞC′ ow, and its role in turbulence generation.

<Corresponding corrections have been done on page 1, lines 26, 27 and on page 2, lines 1-6 of the revised manuscript.>

Page 2 Line 15 "340 m from Baikal ". From what point on the lake coast was the distance counted? Indicate this distance on Figure 2.

<This figure (Fig. 4 of revised manuscript) has been modified.>

Lines 23-24 "The wind in the atmospheric surface layer during the measurements was mostly directed from the north through the mountainous terrain toward Lake Baikal". The sentence arises the following questions: What is your definition of the atmospheric surface layer here? How far the second mountain hill was from the lidar location (Fig.2)? May it influence the measurements of the northerly winds?

<The sentence "The wind in the atmospheric surface . . . toward Lake Baikal" has been removed from the revised text. Description of the terrain has been added on page 5, lines 8-10.>

Line 28 "250 and 750 m ". You may add AGL Line 31 "They are especially marked in the period from 13:30 to 15:30". Suggest to use the word "evident" instead of "marked"

<Fixed, page 5, lines 14 and 18 of the revised manuscript.>

Page 3 Line 11 "Model (1), (2) was applied." Rewrite as "Models (1) and (2) were applied..."

<Fixed, page 6, line 1 of the revised manuscript.>

Summary Page 5, Line 2 "day tame", change to "day time" Suggest to rewrite the next sentence "The low level jet streams were observed by day and night while none of the AIWs events was in the night time," as follows: "The low level jet streams were observed during day and night times while none of the AIWs events were observed in the night time"

<Fixed, page 8, lines 6,7 of the revised manuscript.>

Anonymous Referee #2

This paper presents a few case studies of atmospheric internal waves near the shore of Lake Baikal as well as methodology to infer some wave properties (amplitude, frequency) from the wind profiles obtained with a scanning Doppler lidar. I find the presented case studies and methodology interesting enough to merit publication in AMT when following comments have been addressed.

Major comments

First of all, the structure of this manuscript is unclear. Methods are discussed partly in chapter 1 (introduction) and partly in chapter 2 (observations and analysis), which makes it unnecessarily complicated to follow what has been done. Furthermore, additional details need to be given on the methods and measurements. Please add one more chapter on methods (separate from results) containing detailed description of the instrument, scanning protocols, data processing, the site and the calculations used in inferring the wave characteristics. Please see detailed comments below for more specific questions.

<Section 2 "Lidar, measurement strategy, and data processing" with two Tables and two Figures (Fig. 2 and Fig. 3) has been added in the revised manuscript.>

.

My second major comment is that the analysis on the variation of aerosol concentration is neglecting several sources of uncertainty. As it seems to me that this section does

not contribute significantly to the conclusions of this paper, I recommend the authors to leave it out. If, however, this section is included, all sources of uncertainty in the analysis have to be accounted for. Please see detailed comments below for more specific comments.

<We have replaced "variation of aerosol concentration" by "variation of backscatter coefficient".>

Finally, the results are not discussed with respect to previous studies. Examples of a few previous studies on atmospheric gravity waves (AGW) with a couple of different techniques are listed, but no comparison is made to the characteristics of previous AGW observations or theoretical work. As a consequence the implications of this study (with respect to e.g. NWP, wind power or transport of momentum, energy and trace compounds) are not discussed at all. Please summarize current knowledge on atmospheric internal waves in enough detail to place this work in its context and discuss the differences and similarities of the wave properties from this study with previous observations. For instance, the review by Sun et al. (2015) and the recent studies by Roman-Cascon et al. (2015) and Chouza et al. (2016) might provide a starting point.

<We have added a few references concerning studies on atmospheric gravity waves (AGW), including previous lidar observations of AGW. These studies were carried out for different conditions of occurrence of AGW. The experimental campaign on the coast of Lake Baikal in August of 2015 gave us the first experience of observation of AGW. We think that the comparison with the results of previous studies it would be appropriate after further lidar measurements in parallel with measurements of vertical profiles of temperature and parameters of the turbulence by, for example, radisonde, sodar and sonic anemometers.>

Detailed comments

Page 2, line 10: As there are a couple of different versions of the Halo Streamline lidar, the manuscript has to contain at least the following details of the system in use: wavelength, pulse repetition rate, Nyquist velocity, sampling frequency, velocity resolution, minimum range, range resolution, points per range gate, pulse duration, lens diameter, lens divergence, telescope, integration time per ray. Integration time per ray seems to be changing during the campaign (page 2, line 22) and should be given at least for each case study presented here. Also a reference to a detailed description of the Halo lidar (Pearson et al., 2009) would be good to include.

<Now this information can be found in Table 1 and Table 2 of the revised manuscript. Reference (Pearson et al., 2009) has been included.>

Page 2, line 21-22: It seems that the wind profiles in this study were obtained with a vertical azimuth display (VAD) scan at 60 degree elevation angle. If the VAD scan was carried out by stepping the azimuth angle (as page 2, line 4-6 suggests), please give the number of azimuthal angles included in each scan. Please indicate if the lidar was scheduled to do only VAD scans, or if other scan types were included in the measurement protocol. Please give details on the Doppler lidar data processing. Was some constant threshold applied to SNR to discard range gates with too low signal or were low-signal measurements filtered in some other way? Was the SNR checked for the artefact described by Manninen et al. (2016)? How were missing radial wind observations (e.g. discarded due to too low SNR) handled in reconstructing the wind profile (c.f. Päschke et al., 2016)? Please discuss also the uncertainty in the obtained horizontal and vertical winds (see e.g. Päschke et al., 2016).

<These comments were taken into account in the added section 2. During our experiment at the coast of Lake Baikal in 2015 we used only VAD scan at 60 degree elevation angle. Around 80 % of data has been measured when duration of one scan was 36 s. During measurements at 1 min and 2 min gravity waves were not observed. Therefore in Table 2 we have included only scan duration of 36 s. Figures 3, 4, 8 and 9 (numbering of these figures for previous version of the manuscript) have been modified in accordance with the condition given in Eq. (4) of the revised manuscript. Page 6, lines lines 17-26: "To estimate turbulence strength . . . error is approximately 3 times less."

has been added. Reference (Manninen et al., 2016) has been added.>

Page 2, line 12-14 : "The measurements were conducted in August 14-28 of 2015 on the western coast of Lake Baikal (52_N, 105_E) at the territory of Baikal Astrophysical Observatory of the Institute of Solar-Terrestrial Physics SB RAS, Russia." Please give the measurement site location with better accuracy, 0.01 degree latitude/longitude (or better) would be suitable. As the manuscript deals with "coastal-mountain lee waves" (page 2, line 7), please present a topographic map of the area of interest (with measurement location indicated) so that the reader can assess the role of terrain height on the observations. How far off from a mountain top are these waves observed? Is it the same mountain for each wave case? How did you conclude that the waves reported here are mountain lee waves and not, for instance, generated by jets or fronts (e.g. Plougonven and Zhang, 2014)?

<We have replaced "52_N, 105_E" by "51°50'47.17"N, 104°53'31.21"E". Page 5, line 8: "According to Google map, the profile of the relief surface of the earth starting from position of the lidar in direction to North up to 30 km from the lidar has 10 local maxima with heights of 180 – 420 m and the same number of minimums with heights of 60 – 250 m above Lake Baikal level." has been added. We think that any interested reader can see a topographic map by using the Internet. Because we measured near mountains, we assume that the observed waves are the coastal-mountain lee waves. Unfortunately, we can not prove it without additional experimental research.>

Page 3, lines 9-15: "Model (1), (2) was applied in the analysis of data in Fig.3 for a height of 766.4 m and 47-min time interval starting from 14:20. From these data, with allowance made for the linear trend, we found the wave addends Va(r,t) for the three components of the wind velocity vector. In Fig. 6, the solid curve shows the dependence of Vx on t. To determine the wave frequency fv =1/Tv, we have used experimental function Vx(t) and calculated the spectral density, which is depicted in Fig.7." Please state exactly how the model was applied to the data. For instance: - Is <Va> average (mean or median?) of the u,v,w retrieved from the VAD over the wave

period? If yes, was this period determined visually or with some other method? - What do you mean by "with allowance made for the linear trend" – which parameter has a linear trend and how is it taken into account? How do eq. (1) and (2) look like with this linear trend included? - How were the time-independent parameters in eq. (2) determined – based on a least squares fit? - How was the spectral density of Vx(t) in Fig. 7 calculated? Please provide also some measure of uncertainty for the parameters in eq. (2). For instance bootstrap might be applicable.

<We assume that the mean velocity <Va> is a linear function of time within the time period of 47 min. To remove the linear trend and to obtain the spectral density, we used well-known procedures. To obtain the spectral density with high frequency resolution we added the corresponding number of zeros to the array of the wave addend Vx(t) and then we applied FFT. After determination of the wave frequency (period) from the spectral density, we obtained the amplitude and phase of the wave by well-known least square method (we find the minimum point of the function of the amplitude and phase; this function is the sum of the squares of the difference of the model and measured wave addends). >

Page 4, lines 4-6: "It is known [Smalikho and Banakh, 2015b] that SNR is proportional to the aerosol backscattering coefficient beta = rho sigma, where rho is the concentration of atmospheric aerosol, and sigma is the mean aerosol backscattering cross section." SNR does depend on the aerosol backscattering cross section and concentration, but it depends also on the extinction along the path of the transmitted light (Halo output is attenuated backscatter) and the lidar telescope focus. Furthermore, backscattering cross section depends strongly on relative humidity because of hygroscopic growth of the particles. If this section is included in the final paper the uncertainties from these sources need to be quantified. Please see also Engelmann et al. (2008) for further discussion on aerosol flux measurements using lidar technology. Page 4, lines 6-8: "If we assume that during the measurement time (45 min) SNR does not vary due to turbulent pulsations of the refractive index of air and sigma remains unchanged, then

variations of the aerosol concentration can be estimated from the lidar data as SNR variations." This is probably a good assumption in a well-mixed boundary layer (c.f. Engelmann et al., 2008). However, in this case the aerosol layers are not mixed and the measurement at a constant height can represent quite different aerosol characteristics especially if there was (fresh?) biomass burning smoke present (page 2, line 24). In fact, for the 220 m elevation presented in Fig. 10, the highest SNR seem to coincide with 270 degree wind direction, while lower SNR values are observed for 0-90 degree wind direction (see Fig. 9b). Can you assume that aerosol from opposite wind directions has unchanged optical properties? If not, how does this affect the uncertainty in eq. (3)? Page 4, line 12-14: "Since the SNR oscillates within the height range 100-500 m in Fig. 9(d), it is evident, that aerosol concentration should vary with time too. These aerosol concentration (SNR) variations can be caused by oscillations of the vertical component of the wind velocity vector, whose amplitude is relatively high." Please see previous comments: SNR is not equal to aerosol concentration.

<The text on page 4 (lines 4-20) "It is known ... by about 90 deg." in the previous version of the manuscript has been replaced by the text on page 7, lines 3-23 of the revised manuscript).>

Minor comments

Please check the language carefully once more.

Page 1, line 8-9: Please specify if height is above ground level or something else

<Fixed, page 1, line 11.>

Page 1, line 21: "Xiaofeng et al., 2001" is not included in References

<"Xiaofeng" has been replaced by "Li".>

Page 2, line 4-5: "During the measurements, the elevation angle phi is fixed, while the azimuth angle theta = omega t of the beam axis position varies with time t and with the rate omega." To me this statement suggests that the VAD was operated with "continuous wave" method, though the Halo lidar is more commonly operated by stepping from one azimuth angle to the next, in which case scan duration is determined by the integration time and number of azimuthal directions included in the scan. Please clarify how the VAD scan was operated. Page 2, line 6: "k,n=1,2,3,. . ." Please give the number of range gates and azimuth angles used in this study.

<It is done in Section 2 "Lidar, measurement strategy, and data processing" of the revised manuscript.>

Page 2, line 7-9: "Lee AIWs (or orographic waves) are one of the types of AGWs, which arise on leeward of obstacles at the stable stratification of an incoming flow [Vel'tishchev and Stepanenko, 2006; Kozhevnikov, 1999]." There are many earlier studies on lee waves. Some early references can be found e.g. in Sun et al. (2015) and in Makarenko and Maltseva (2011).

<References (Makarenko and Maltseva, 2011) and (Sun et al., 2015) have been included.>

Page 2, line 15: Is the height above Lake Baikal level?

<Page 5, line 8 (in the revised manuscript): "above the lake level" has been added.>

Page 2, line 23: "The wind in the atmospheric surface layer during the measurements was mostly directed from the north through the mountainous terrain toward Lake Baikal." What height above ground is "the atmospheric surface layer"? Is this from the Doppler lidar or some surface anemometer?

<This sentence has been deleted.>

Page 3, line 1: "Neglecting the wind turbulence, we use the model of a plane wave for the component of the wind velocity vector" Please discuss the uncertainty in neglecting turbulence.

<We try to do it on page 6, lines 17-26 of the revised manuscript.>

Page 3, line 5: What is the coordinate system for the radius vector?

<Page 5, line 24 (of the revised manuscript): "in the Cartesian system of coordinates with center at point of the lidar position" has been added.>

Page 3, line 8-9: "If the wind direction coincides with the direction of propagation of the internal wave, . . ." How is the wavelength determined if the wind direction does not coincide with the direction of propagation of the internal wave?

<From our lidar measurements we do not determine the wavelength of the internal wave, we can estimate only the time period of the internal wave that is the same for all component of the wind vector (amplitudes and phases may differ).>

Page 3, line 24-25: "Since the amplitude A is not 0 (see Fig. 3(b) and Fig. 5(b)), the direction of propagation of the internal wave did not coincide with the wind direction." How large was the angle between wind direction and the direction of propagation of the internal wave (for each wave case)? How does this compare with previous observations?

<We did not determine the direction of propagation of the internal wave.>

Page 3, line 28-29: "According to the data of Fig. 8(b), the period and amplitude of the wave were, respectively, 9 min and 0.9 m/s." Is this for horizontal wind speed?

<Figure 10 (in the revised manuscript) shows the wind velocity (or horizontal wind speed).>

Page 4, line 4-5: "It is known [Smalikho and Banakh, 2015b] that SNR is proportional to the aerosol backscattering coefficient" Here a reference to e.g. the instrument description (Pearson et al., 2009) might be more appropriate.

<We have removed "[Smalikho and Banakh, 2015b]".>

Page 4, line 22: Is time local time?

<Everywhere in the text, we have added "LT".>

Page 5, line 5-6: "A total of six cases of AIW formation have been revealed, which always occurred in presence of one or two (in 5 of 6 cases) narrow jet streams at heights of about 200 and 700 m." I find interesting the observation of two simultaneous jet streams. Are there previous observations of such cases from comparable environments? Can you comment on the possible forcing mechanisms of the jet streams?

<We observed presence of two narrow jets in AB for the first time and have no explanation of this phenomenon.>

Figure 1. It seems a little unusual to me to define VAD with a left-handed axis. Is this the axis used for eq. (1) and (2)?

<Yes.>

Figure 8. Please use the same colour map as for other wind speed plots.

<Fixed, Fig 10 of the revised manuscript.>

References Chouza, F., Reitebuch, O., Jähn, M., Rahm, S. and Weinzierl, B.: Vertical wind retrieved by airborne lidar and analysis of island induced gravity waves in combination with numerical models and in situ particle measurements, Atmos. Chem. Phys., 16(7), 4675–4692, doi:10.5194/acp-16-4675-2016, 2016. Engelmann, R., Wandinger, U., Ansmann, A., Müller, D., Žeromskis, E., Althausen, D. and Wehner, B.: Lidar Observations of the Vertical Aerosol Flux in the Planetary Boundary Layer, Journal of Atmospheric and Oceanic Technology, 25(8), 1296–1306, 2008. Makarenko, N. I. and Maltseva, J. L.: Interference of lee waves over mountain ranges, Nat. Hazards Earth Syst. Sci., 11(1), 27–32, doi:10.5194/nhess-11-27-2011, 2011. Manninen, A. J., O'Connor, E. J., Vakkari, V. and Petäjä, T.: A generalised background correction algorithm for a Halo Doppler lidar and its application to data from Finland, Atmos. Meas. Tech., 9(2), 817–827, doi:10.5194/amt-9-817-2016, 2016. Pearson, G., Davies, F. and Collier, C.: An Analysis of the Performance of the UFAM Pulsed

Doppler Lidar for Observing the Boundary Layer, J. Atmos. Oceanic Technol., 26(2), 240–250, doi:10.1175/2008JTECHA1128.1, 2009. Päschke, E., Leinweber, R., and Lehmann, V.: An assessment of the performance of a 1.5 $\mu$m Doppler lidar for operational vertical wind profiling based on a 1-year trial, Atmos. Meas. Tech., 8, 2251–2266, doi:10.5194/amt-8-2251-2015, 2015. Plougonven, R. and Zhang, F.: Internal gravity waves from atmospheric jets and fronts, Reviews of Geophysics, 52, 33-76, doi:10.1002/2012RG000419, 2014. Román-Cascón, C., Yagüe, C., Mahrt, L., Sastre, M., Steeneveld, G.-J., Pardyjak, E., van de Boer, A. and Hartogensis, O.: Interactions among drainage flows, gravity waves and turbulence: a BLLAST case study, Atmos. Chem. Phys., 15(15), 9031–9047, doi:10.5194/acp-15-9031-2015, 2015. Sun, J., Nappo, C. J., Mahrt, L., Belušiʹc, D., Grisogono, B., Stauffer, D. R., Pulido, M., Staquet, C., Jiang, Q., Pouquet, A., Yagüe, C., Galperin, B., Smith, R. B., Finnigan, J. J., Mayor, S. D., Svensson, G., Grachev, A. A. and Neff, W. D.: Review of wave-turbulence interactions in the stable atmospheric boundary layer, Reviews of Geophysics, 53. 956-993, doi:10.1002/2015RG000487, 2015.

Please also note the supplement to this comment:
http://www.atmos-meas-tech-discuss.net/amt-2016-165/amt-2016-165-AC1-supplement.zip